# Association between Statin Use and Gastric Cancer: A Nested Case-Control Study Using a National Health Screening Cohort in Korea

**DOI:** 10.3390/ph14121283

**Published:** 2021-12-08

**Authors:** Mi Jung Kwon, Ho Suk Kang, Joo-Hee Kim, Ji Hee Kim, Se Hoon Kim, Nan Young Kim, Eun Sook Nam, Kyueng-Whan Min, Hyo Geun Choi

**Affiliations:** 1Department of Pathology, Hallym University Sacred Heart Hospital, Hallym University College of Medicine, Anyang 14068, Korea; mulank@hanmail.net; 2Division of Gastroenterology, Department of Internal Medicine, Hallym University Sacred Heart Hospital, Hallym University College of Medicine, Anyang 14068, Korea; hskang76@hallym.or.kr; 3Division of Pulmonary, Allergy, and Critical Care Medicine, Department of Medicine, Hallym University Sacred Heart Hospital, Hallym University College of Medicine, Anyang 14068, Korea; luxjhee@gmail.com; 4Department of Neurosurgery, Hallym University Sacred Heart Hospital, Hallym University College of Medicine, Anyang 14068, Korea; kimjihee.ns@gmail.com; 5Department of Pathology, Severance Hospital, Yonsei University College of Medicine, Seoul 03722, Korea; PAXCO@yuhs.ac; 6Hallym Institute of Translational Genomics and Bioinformatics, Hallym University Medical Center, Anyang 14068, Korea; honeyny78@gmail.com; 7Department of Pathology, Kangdong Sacred Heart Hospital, Hallym University College of Medicine, Seoul 05355, Korea; esnam@kdh.or.kr; 8Department of Pathology, Hanyang University Guri Hospital, Hanyang University College of Medicine, GurI 11923, Korea; kyueng@gmail.com; 9Department of Otorhinolaryngology-Head & Neck Surgery, Hallym University Sacred Heart Hospital, Hallym University College of Medicine, Anyang 14068, Korea

**Keywords:** gastric cancer, statin, nested case–control study, Nationwide Health Insurance research database

## Abstract

Concerns about the hazards of statins on the development and mortality of stomach cancers remain controversial. Here, we investigated the likelihood of incident gastric cancers and related mortality depending on statin exposure, statin type, and the duration of use. This nested case–control-designed study was composed of 8798 patients who were diagnosed with gastric cancer and matched with 35,192 controls at a 1:4 ratio based on propensity scores of age, sex, residential area, and income from the Korean National Health Insurance Service—Health Screening Cohort database (2002–2015). Propensity score overlap weighting was adjusted to balance the baseline covariates. Overlap propensity score-weighted logistic regression analyses were assessed to determine associations of the prior use of statins (any statin, hydrophilic statins vs. lipophilic statins) with incident gastric cancer and its mortality depending on the medication duration (<180 days, 180–545 days, and >545 days) after adjusting for multiple covariates. After adjustment, the use of any statin, hydrophilic statins, or lipophilic statins showed significant associations with lower odds for incident stomach cancer when used for a short-term period (180–545 days) (OR = 0.88, 95% CI = 0.81–0.86, *p* = 0.002; OR = 0.78, 95% CI = 0.66–0.92, *p* = 0.004; and OR = 0.91, 95% CI = 0.84–0.99, *p* = 0.039, respectively) compared to the control group. Hydrophilic statin use for 180–545 days was associated with 53% lower overall mortality (OR = 0.47; 95% CI = 0.29–0.77). In subgroup analyses, beneficial effects on both cancer development and mortality persisted in patients ≥65 years old, patients with normal blood pressure, and patients with high fasting glucose levels. There were no such associations with long-term statin use (>545 days). Thus, the current nationwide cohort study suggests that prior short-term statin use may have anti-gastric cancer benefits in elderly patients with hyperglycemia.

## 1. Introduction

Stomach cancers are the fifth major cause of both cancer and cancer-related death in Korea, one of the geographical regions with the highest risk for gastric cancer worldwide [1]. The important risk factors for gastric cancer include advanced age, male sex, family history, *Helicobacter pylori*, history of chronic atrophic gastritis or pernicious anemia, obesity, chemical carcinogen use, smoking, red meat, alcohol consumption, and low socioeconomic status [2]. Over the last two decades, it is clear that mandatory national endoscopic screening programs for gastric cancer in adults over 40 years of age every two years have reduced mortality from gastric cancer [3]. Nevertheless, gastric cancer is still the most commonly diagnosed cancer in adult men, and the incidence of gastric cancer increases steadily with age for both men and women in Korea [1], indicating that screening per se does not entirely enable control of the incidence of stomach cancer; hence, determining the potential risk factors for stomach cancers is necessary for fundamental prevention.

Statins are the most commonly prescribed lipid-lowering drug agent for treating hypercholesterolemia and cardiovascular disease and act as competitive inhibitors of 3-hydroxy-3-methylglutaryl-coenzyme A (HMG-CoA) reductase, the rate-limiting enzyme that regulates the synthesis of cholesterol in the mevalonate pathway [4]. Statins also have noncardiovascular supplementary anti-inflammatory, antioxidative, and immunomodulatory properties [4]. Particularly in terms of cancer, the role of statins is still controversial regarding their carcinogenic or anticancer effects [5,6,7], which raised safety concerns of statin use on the risk of cancers. Prospective data from clinical trials suggest that statins actually increase the incidences of breast cancer and prostate cancer long after over 15 follow-up years of statin treatment, possibly due to a statin-induced increase in regulatory T cells, resulting in impaired host antitumor immunity [8,9,10]. Statin types, according to hydrophobicity properties (hydrophilic or lipophilic), may influence antiproliferative effects in cancers. Due to the high hepatic washout selectivity of hydrophilic statins and the high penetration ability into the cell plasma membrane of lipophilic statins, lipophilic statins may be effective in nonhepatic, nonintestinal, solid organ cancers, including breast and ovary cancers [11,12,13]. Nevertheless, statins have drawn attention for their potential anticancer effects in gastric cancers among Asian countries with a high prevalence of gastric cancers [14,15,16,17,18]. In vitro evidence indicates the anticancer activity of statins in gastric cancers, primarily by means of downregulation of the mevalonate pathway, which is critical for vital cellular functions maintaining membrane integrity and cell cycle progression [5,6,18,19,20]. Statins have been reported to synergistically reduce tumor growth in conjunction with radiation or chemotherapy by increasing radiosensitization or chemosensitization through the induction of apoptosis in experimental gastric cancer animal models [18,21]. Population-based studies and meta-analyses have supported the reduced gastric cancer risk in individuals who had received statins [17,22,23]. There are two nationwide epidemiological studies regarding the influence of prior statin use on the development of stomach cancers based on Korean individuals [16,24]. These studies did not specifically investigate the relationships of the duration of statin use or statin types with the incidence and mortality of gastric cancer, and whether prior statin use affects the risk or mortality of gastric cancer according to the statin types and duration of use remains debated. In contrast, one Dutch study dealing with pharmacy databases and other meta-analyses conducted in Japan denied that gastric cancer risk is affected by statin use [25,26]. Hence, statins may have advantageous impacts in lowering stomach cancer risk, but their clinical relevance remains obscure in prospective studies and clinical trials [27,28].

We hypothesized that there would be certain risk factors related to statin use that would be predictive of incident gastric cancer and its prognosis. This study extended previous studies in the field; we further investigated potential risk factors related to statin use that may be predictive of incident gastric cancer and its mortality according to statin type and duration of use. To explore this, a nationwide cohort study with an exactly matched nested case–control design was conducted, together with comprehensive subgroup analyses, to estimate the potential impacts of statins on the incidence and mortality of subsequent gastric cancers.

## 2. Results

### 2.1. Baseline Characteristics of the Study Participants

A total of 8798 patients with gastric cancer and 35,192 comparison participants were enrolled in the present study after propensity score matching. Before adjustment using the overlap weighting method, the baseline characteristics between the groups were not identical in terms of dyslipidemia, obesity status, fasting blood glucose levels, blood pressure, smoking status, total cholesterol levels, Charlson Comorbidity Index (CCI) scores, or alcohol consumption. After overlap weighting adjustment for the baseline covariate imbalances, the standardized mean differences were minimized, and the balance between the groups was exactly identical (standardized difference = 0.00) (Table 1).

### 2.2. Odds Ratios of the Incidence of Gastric Cancer for the Duration of Use and Types of Statins

We examined the odds of incident gastric cancers according to the designated periods of either using any statin or the statin type (hydrophilic vs. lipophilic) (Table 2). The use of any statin, hydrophilic statins, or lipophilic statins was related to reduced odds for incident gastric cancers when used for a short-term period (180–545 days) (odds ratio (OR) = 0.88, 95% confidence interval (CI) = 0.81–0.86, *p* = 0.002; OR = 0.78, 95% CI = 0.66–0.92, *p* = 0.004; and OR = 0.91, 95% CI = 0.84–0.99, *p* = 0.039, respectively). Long-term statin use (>545 days) showed no statistical association with incident gastric cancer.

We comprehensively performed subgroup analyses according to 29 baseline covariates. Overall, the subgroup analyses for any statin, hydrophilic statins, and lipophilic statins supported the association of statin use for 180–545 days with a reduced risk of gastric cancer, showing either overlapping or their own specific risk factors for gastric cancer. The significance shown in any statin users was clear in the following subgroups: patients aged over 65 years (0.81; 95% CI = 0.73–0.91; *p* < 0.001); men (0.89; 95% CI = 0.80–0.98; *p* = 0.021); patients with high incomes (0.79; 95% CI = 0.71–0.88; *p* < 0.001); rural residents (0.86; 95% CI = 0.77–0.96; *p* = 0.006); obese patients (0.82; 95% CI = 0.72–0.93; *p* = 0.002); nonsmoking patients (0.86; 95% CI = 0.78–0.96; *p* = 0.005); patients with frequent alcohol consumption (≥1 time/week) (0.84; 95% CI = 0.74–0.96; *p* = 0.008); patients with high total cholesterol levels (≥200 mg/dL) (0.83; 95% CI = 0.74–0.94; *p* = 0.002); patients with normal blood pressure (diastolic blood pressure (DBP) < 90 mmHg and systolic blood pressure (SBP) < 140 mmHg) (0.86; 95% CI = 0.77–0.95; *p* = 0.002); patients with a CCI score of 1 (0.79; 95% CI = 0.66–0.96; *p* = 0.019); patients with high fasting blood glucose levels (≥100 mg/dL) (0.84; 95% CI = 0.75–0.94; *p* = 0.003); patients with CCI scores ≥2 (0.84; 95% CI = 0.71–0.98; *p* = 0.030); and patients with nondyslipidemia (0.65; 95% CI = 0.52–0.80; *p* < 0.001) (Figure 1a,b and Appendix A).

Common significant subgroups across any statin, hydrophilic statins, and lipophilic statins included patients aged over 65, obese patients, patients with high fasting blood glucose levels, patients with nondyslipidemia, and patients with normal blood pressure, which showed significantly reduced odds for gastric cancer.

Specific significant subgroups exclusively for hydrophilic statins or lipophilic statins were patients with low incomes and normal weight patients for hydrophilic statins (Figure 2a,b and Appendix A) and urban residents for lipophilic statins (Figure 3a,b and Appendix A), which showed reduced odds for gastric cancer.

### 2.3. Odds Ratios of Mortality in Gastric Cancer Patients for the Duration of Use and Types of Statins

Imbalanced baseline characteristics between the deceased and surviving participants with gastric cancer who had ever used statins were further adjusted using overlap weighting to equally balance the groups (standardized difference = 0.00) (Table 3).

After propensity score overlap weighting, the short-term use of hydrophilic statins (180–545 days) significantly reduced mortality only in the cohort with stomach cancer who had been exposed to statins (0.47; 95% CI = 0.29–0.77; *p* = 0.003) (Table 4). This inverse association remained valid in the subgroups of patients aged over 65, patients with high fasting glucose or total cholesterol levels, patients with dyslipidemia, men, overweight patients, patients who underwent surgery, nonsmokers, urban residents, patients with high incomes, and patients with normal blood pressure (Figure 4a,b and Appendix A).

Conversely, we noted certain associations with an increase in mortality through comprehensive subgroup analyses. Mortality was increased in subgroups of rural residents with long-term hydrophilic statin use (>545 days) (1.91; 95% CI = 1.04–3.52; *p* = 0.037), women, nonsmokers, patients with frequent alcohol consumption, patients with high total cholesterol levels, and patients with a CCI score of 0 for short-term use of any statin (180–545 days). Mortality was also increased for women, nonsmokers, patients with high incomes, normal weight patients, patients with frequent alcohol consumption, patients with high total cholesterol levels, patients with a CCI score of 0, patients with dyslipidemia, and patients who underwent surgery with adjuvant therapy with short-term lipophilic statin use (180–545 days) (Appendix A for any statin; Appendix A for lipophilic statins).

## 3. Discussion

In this large nationwide cohort study, we demonstrated that prior short-term statin use reduced the likelihood of gastric cancer and its overall mortality regardless of statin type. The anticancer effect of statins in both reduced risk for gastric cancer and mortality was found in in patients aged over 65, patients with normal blood pressure, and patients with high fasting glucose levels. Since gastric cancer accounts for one of the prevalent malignancies in elderly patients aged over 65 in Korea [1], the most noteworthy finding is that statin use has reduced the development and mortality of a subset of gastric cancers in elderly individuals suffering from hyperglycemia.

Earlier studies conducted in the Netherlands [25] and Denmark [7] claimed no relation of statin use to gastric cancer risk. These studies included a series of various cancer types with only a limited sample size of gastric cancers and did not specifically focus on gastric cancers. In the last two decades, there have been increasing studies to support the preventive effect of statins in gastric cancer. We also observed that the prior use of any statin was related to a 12% reduction in the development of stomach cancers in the short-term period (180–545 days) (OR = 0.88; 95% CI = 0.81–0.86). This result was partially compatible with that of population-based research conducted in Taiwan, which demonstrated that using any statin markedly reduced stomach cancer risk (OR = 0.68, 95% CI = 0.49–0.95) [29]. A meta-analysis encompassing observational literature and randomized controlled trials also confirmed the anticancer effect of statins, with a 27% reduction in the risk for stomach cancer (95% CI = 0.58–0.93) [12]. Interestingly, one case–control study conducted in a single hospital noted a much lower OR of 0.211 (95% CI = 0.159–0.281) for stomach cancer in diabetic patients who had received statin therapy than in patients with other cancer types [17]. This risk decrease of up to 80% for gastric cancer associated with prior statin use in diabetic patients was seemingly considered due to bias by the authors [17]. However, this decreased OR corresponds with our findings that high fasting blood glucose levels in short-term statin users were an independent favorable factor associated with a reduced risk of developing stomach cancer.

The protective relevance of statins for mortality reduction in our study corresponds with the results from English and Scottish population-based studies. One of these studies indicated a relationship between prior statin use and decreased mortality (hazard ratio = 0.91; 95% CI = 0.84–0.98) [30], which showed a much lower reduction in mortality (9%) than the result of 53% reduced mortality in our study (OR = 0.47; 95% CI = 0.29–0.77). The difference may be attributable to the different epidemiological cohorts and ethnicities, in which there may be predisposing factors that make patients susceptible to gastric cancer. We approximated the estimated duration to obtain the beneficial effect of statins. Notably, the reduced odds regarding cancer incidence and mortality were identified as unique in the short-term duration (180–545 days) of statin use. No such associations were found for long-term statin use (>545 days). A meta-analysis dealing with case–control studies also demonstrated no correlation between the cumulative use duration of statins and gastric cancer risk across global populations in Asia, Europe, or the USA [23]. Considerable heterogeneity in categorizing the duration of use among studies may contribute to inconclusive results, with some protective effects in prolonged statin use status before a gastric cancer diagnosis > 6 months [17], >4 years [25], or ≥5 years [22]. The limited duration and response between gastric cancer and statins in our study might imply that the minimal duration required to reach any threshold of pleiotropic effects would suffice to enhance apoptosis and block the growth of malignant gastric cells [7,18]. The result would be comparable to that for simvastatin, which at a concentration of 4 μmol/L for 1 day induces gastric cancer cells to undergo apoptotic cell death [5].

Information regarding the association between statins and the likelihood of stomach cancer according to statin properties is very limited. It appears that their effects on cancers are highly dependent on the physiochemical properties of statins in esophageal, lung, and colorectal cancers, and in brain tumors [31]. However, the anticancer efficacy according to the properties of different statin types has scarcely been described in the case of gastric cancer. In this study, hydrophilic statins (OR = 0.78; 95% CI = 0.66–0.92) and lipophilic statins (OR = 0.91; 95% CI = 0.84–0.99) used for a short-term period (180–545 days) were associated with a negative impacts on the development of gastric cancer, indicating that both hydrophilic and lipophilic statins seem to have preventive effects on gastric cancer. In vitro studies of lipophilic lovastatin and simvastatin have shown that they suppress proliferation, migration, and invasion, and promote apoptosis of human gastric malignant cells [6,21]. There has been little research regarding gastric cancer cell lines applying hydrophilic statins. In our study, a remarkable reduction in overall mortality was only observed among the patients previously using hydrophilic statins. The possible beneficial impact of hydrophilic statins has been provided as a clue with a low hazard ratio of 0.70 on overall survival in patients with ovarian cancers [11]. Meta-analysis has also shown improved all-cause mortality in breast cancer patients who had ever used hydrophilic statins [13]. However, two multicenter studies and randomized phase II trials failed to demonstrate beneficial effects of a hydrophilic pravastatin additive to standard chemotherapy in reducing death or recurrent events after surgical resection of advanced-stage stomach cancers [27,28]. On the other hand, the subsequent use of statins in postendoscopic resection of early gastric cancers has shown a significant decline in the occurrence of metachronous recurrences (hazard ratio = 0.17; 95% CI 0.13–0.24) [32]. The discrepancy with our results may be because the former two studies enrolled specific subgroups of gastric cancers with a worse prognosis and advanced stage with a small cohort (30–60 patients), and all patients were evaluated for postsurgical statin use. Statin therapy might be rather effective in the early diagnosis of gastric cancer. Lovastatin strongly suppresses the expression of genes involved in cell division in gastric cancer cells and exhibits cytotoxic synergy with platinum treatment, triggering efficient apoptosis in early gastric cancer cells [21]. Primary gastric cancer cells are sensitive to antitumor drugs, including statins, whereas metastatic stomach cancer cells are highly resistant to these drugs, with certain different molecular mechanisms of cholesterol absorption operating between primary and metastatic gastric malignant cells [5].

The mechanism underlying the relationship of statin use with a decreased risk of gastric cancer and death caused by gastric cancer remains unclear. Since HMG-CoA reductase is upregulated in gastric cancer, promoting tumor growth and migration [20], the blockade of HMG-CoA reductase by statins has been considered to be the key mechanism involved in the chemoprevention of statins in gastric cancer [20]. As downstream products in the mevalonate signaling pathway critically contribute to pivotal cellular functions that maintain membranous integrity, signaling pathways, protein synthesis, and cell cycle progression, the downregulation of the mevalonate pathway by statins triggers apoptosis and reduces growth by inhibiting cell cycle progression and major cellular functions in cancer cells and by increasing radiosensitization or chemosensitization [18,20]. These alterations in gastric cancer cell metabolism might consequently lead to clinical outcomes in patients with gastric cancer [5].

Our current epidemiological study updated the significant impact of prior statin and hydrophilic statin use for a limited period in terms of the decreased incidence and mortality of gastric cancer, which highlights our study extension, distinguishable from two previous Korean National Health Insurance Service—Health Screening (KNHIS-HS)-based studies regarding prior statin use and gastric cancer [16,24]. The investigators in these studies did not perform a comprehensive subgroup analysis of the patients to explore the potential risk factors for gastric cancer and its mortality [16,24]. Our study had a more expanded scope, extracted pooled data from 2002 to 2015, and further analyzed the risk difference of gastric cancer using more detailed information about the duration of statin use and the types of statins used. Our study is noteworthy for a homogenous and exactly balanced cohort including gastric cancer patients and control participants using overlap weighting adjustments.

The strengths of the present study are based on a representative, nationwide cohort database with a balance of patients and control participants, which makes our findings more generalizable. Since the KNHIS-HS data include all the hospitals and clinics of the nation with no exceptions, no medical history was lost in the follow-up. We comprehensively considered possible confounders. The exactly balanced adjustments of socioeconomic status and potential risk factors and comorbidities related to gastric cancer or statin users (e.g., total cholesterol level, alcohol consumption, smoking status, blood pressure, obesity status, and fasting blood glucose levels) are additional strengths.

The current study does, however, have limitations that should be addressed. As this study registered patients based on diagnosis codes and included only Korean subjects, unmeasured confounding effects could not be completely excluded. Since we used prescription dates, patient compliance could not be confirmed. Statin-users were treated with different statins. No information pertaining to *Helicobacter pylori*, stage, histology and differentiation, dosage and frequency of statin use, family history, and genetic data concerning gastric cancer or related systemic diseases was included in the health insurance database, so the possibility of missing data was not taken into consideration.

In summary, our results indicate that prior short-term use of statins may have clinical significance in terms of a decreased likelihood of gastric cancer and mortality onset, especially in elderly patients with high fasting glucose levels. This nationwide cohort study, using Korean insurance claims data, provides supporting evidence from previous studies which demonstrated the anticancer benefits of statins against gastric cancers and their mortality.

## 4. Materials and Methods

### 4.1. Ethical Approval and Study Population

This study was approved by the Institutional Review Board at Hallym University Sacred-Heart Hospital (IRB No. 2019-10-023), with exempted written informed consent. The study proceeded in accordance with the regulations and guidelines of the Institutional Ethics Committee.

The study was conducted with data retrieved from the KNHIS-HS cohort, which offers population-based electronic files on the Korean population for research purposes, deidentified, with anonymous information substituted for identification codes, as previously described [33]. The diagnostic codes of the KNHIS-HS data follow the International Classification of Diseases, 10th Revision, Clinical Modification (ICD-10-CM).

Since a nested case–control study design is suitable to retrospectively identify causal associations of the history of subjects for the presence or absence of an exposure in outcome status at the outset of the investigation [34], we used a nested case–control design for the study. Patients who were newly diagnosed with gastric cancer and were older than 40 years were initially identified out of 514,866 people with 615,488,428 medical claim codes between 2002 and 2015 (*n* = 11,402). Gastric cancer was defined as the main diagnosis with ICD-10 codes, such as malignant neoplasm of cardia (C16.0), fundus of stomach (C16.1), body of stomach (C16.2), pyloric antrum (C16.3), pylorus (C16.4), lesser curvature of stomach, unspecified (C16.5), greater curvature of stomach, unspecified (C16.6), overlapping sites of stomach (C16.8), and malignant neoplasm of stomach, unspecified (C16.9). Among these assigned tumor codes, we only included participants with more than three clinic visits who were treated with surgical resection for the reliability of inclusion to reduce the possibility of false positives for stomach cancer. We excluded patients if they were diagnosed in the period from 2002–2004 (3-year wash-out period, *n* = 2602) and did not have records for body mass index (BMI) or fasting blood glucose results (*n* = 2). As a result of the exclusion of patients diagnosed in 3-year wash-out periods, the final gastric cancer group only included patients diagnosed on 1 January 2005 or later, and their earliest time for follow-up in the study was 1 January 2005.

Comparisons of individuals who had never been diagnosed with stomach cancers before 2002 were initially extracted from the database (*n* = 503,464), using random number generation to reduce possible selection bias. We excluded any participants if they were diagnosed with other malignant neoplasms of digestive organs (ICD-10 codes: C15–C26), with more than two clinic visits with assigned codes (*n* = 23,806), such as malignant neoplasms of the esophagus (C15), small intestine (C17), colon (C18), rectosigmoid junction (C19), rectum (C20), anus and anal canal (C21), liver and intrahepatic bile ducts (C22), gallbladder (C23), other and unspecified parts of the biliary tract (C24), pancreas (C25), and other and ill-defined digestive organs (C26), or participants who were diagnosed with stomach cancer (ICD-10 code: J16) with fewer than three clinic visits with assigned codes (*n* = 1500).

To minimize the differences in the gastric cancer and comparison groups’ baseline demographic and clinical characteristics, propensity score matching was performed based on age, sex, income, and area of residence. Using this method, participants with gastric cancer were individually matched with control participants based on similar propensity score values. The index date of every gastric cancer patient was defined as the day when the ICD-10 codes for stomach cancer (C16.0–C16.9) were electronically assigned to the patients in the health insurance claims datasets. The index date of the comparison group was defined as the index date of their matched gastric cancer patient. Therefore, each matched patient and comparison participant group had the same index date. During the matching process, 442,966 comparison participants were excluded. Finally, a total of 8798 gastric cancer patients were matched with 35,192 control participants at a 1:4 ratio (Figure 5). Then, we searched for a previous history of statin use before the occurrence of gastric cancer and subsequent patient survival outcomes.

### 4.2. Exposure (Statin)

Participants who first used statins within two years before the index date were eligible for the study. The prior statin use duration was measured as the total prescription dates of statins for two years (730 days) before the index date because the effects of statins last for two years [35]; the duration was divided as <180 days, 180–545 days, and >545–730 days, as previously modified [17,25,30]. Statin users were considered patients who had statin prescriptions for a minimum of 180 days [17,25,30]. The patients deemed statin nonusers had prescriptions for <180 days, short-term users had prescriptions from 180–545 days, and long-term users had prescriptions for >545 days, as previously described [17,30]. Persons with any previous statin history more than 2 years from the index date in their medical records were excluded.

The impacts of statin use duration on gastric cancer and patient death were analyzed using conditional logistic regression. Since the statins surveyed in this study included simvastatin, atorvastatin, lovastatin, pravastatin, fluvastatin, and rosuvastatin, they were categorized as lipophilic statins (atorvastatin, simvastatin, lovastatin, and fluvastatin) and hydrophilic statins (pravastatin and rosuvastatin) to investigate potential impacts according to the lipid affinity of the statin types on the occurrence and mortality of gastric cancer.

### 4.3. Outcome (Gastric Cancer)

Gastric cancer was defined according to the ICD-10 codes (C16.0–C16.9) that were assigned with more than three clinic visits. The primary outcomes were the occurrence of gastric cancer with prior statin prescription dates of <180 days, 180–545 days, and >545 days, depending on the statin type. The secondary outcome was the odds for overall mortality of these patients according to the statin type and duration of use.

### 4.4. Covariates

The participants were divided into 10 age groups based on 5-year intervals and 5 income groups (class 1 (lowest income) to class 5 (highest income)). Briefly, the participants were initially divided by income into 41 classes (one health aid class, 20 self-employed health insurance classes, and 20 employed health insurance classes). These groups were then recategorized into five classes (class 1 (lowest income)–class 5 (highest income)) [36]. Region of residence was divided into 16 areas according to administrative district. These regions were stratified based on urban (Seoul, Busan, Daegu, Incheon, Gwangju, Daejeon, and Ulsan) and rural (Gyeonggi, Gangwon, Chungcheongbuk, Chungcheongnam, Jeollabuk, Jeollanam, Gyeongsangbuk, Gyeongsangnam, and Jeju) areas [36]. Obesity status using BMI (kg/m^2^; <18.5 for underweight, ≥18.5 to <23 for normal weight, ≥23 to <25 for overweight, ≥25 to <30 for obese I, ≥30 for obese II), alcohol consumption (<1 time a week, ≥1 time a week), and smoking status (nonsmoker, past smoker, current smoker) were categorized in a similar manner as our previous study [37]. The total cholesterol (mg/dL) level, fasting blood glucose (mg/dL) level, DBP (mmHg), and SBP (mmHg) were calculated. The CCI was summed as a total score from 0–29 to quantify disease burden using 17 comorbidities. The CCI score was calculated without including stomach cancer. Dyslipidemia was determined as the ICD-10 code (E78) with ≥2 clinic visits.

### 4.5. Statistical Analyses

Categorical data were summarized as percentages. Continuous data were depicted as the mean and standard deviation. We conducted propensity score overlap weighting to optimize the covariate balance and an effective sample size. The propensity score was calculated by multivariable logistic regression with the aforementioned four baseline covariates. During propensity score matching, we used a greedy, nearest-neighbor matching algorithm to form pairs of gastric cancer and control participants [38]. Propensity scores where gastric cancer patients and control participants were weighted by the probability of a propensity score of 1 and the probability of a propensity score, respectively, were applied for overlap weighting calculated between 0 and 1 [39,40,41]. To assess bias reduction, we checked the balance of the matched data between the groups in terms of absolute standardized differences of the covariates before and after matching. An absolute standardized difference of <0.20 indicated good balance for a particular covariate [38].

Propensity score overlap weighted multivariable logistic regressions for crude (unadjusted) and overlap weighted (adjusted for all covariates) models were used to estimate overlap weighted ORs and 95% CIs for incident gastric cancer and its mortality with respect to the period of statin use and statin types by adjusting for potential confounders. Subgroup analyses were carried out according to all covariate variables (Appendix A). Two-tailed analyses were used, with a *p*-value less than 0.05 considered statistically significant. SAS version 9.4 (SAS Institute Inc., Cary, NC, USA) was used for all statistical analyses.

## Figures and Tables

**Figure 1 pharmaceuticals-14-01283-f001:**
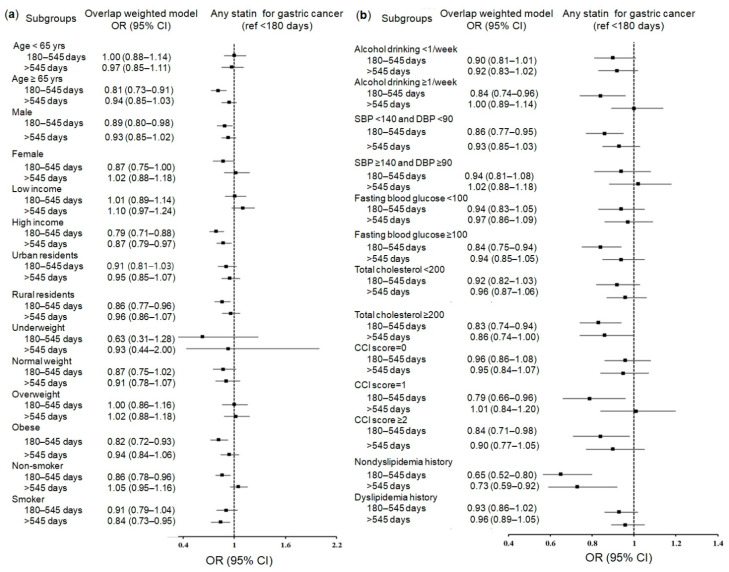
Forest plots for multivariable conditional logistic regression depicting the overlap weighted odds ratios (95% confidence intervals) of previous use duration of any statin for incident gastric cancer according to comprehensive subgroup analyses, including age, sex, income, region of residence, obesity, and smoking (**a**), alcohol consumption, systolic blood pressure, diastolic blood pressure, fasting blood glucose, total cholesterol, CCI scores, and dyslipidemia history (**b**). The reference period is <180 days. Full results of the crude and adjusted overall weighted models are available in Appendix A (any statin).

**Figure 2 pharmaceuticals-14-01283-f002:**
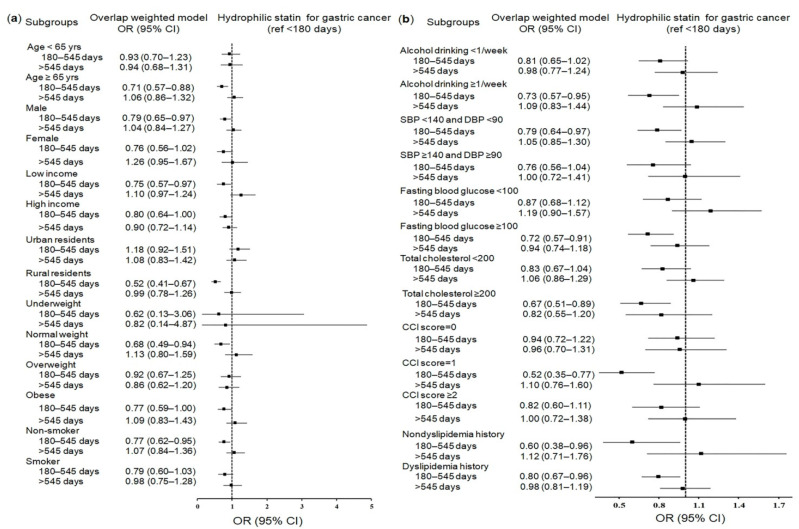
Forest plots for multivariable conditional logistic regression depicting the overlap weighted odds ratios (95% confidence intervals) of previous use duration of hydrophilic statins for incident gastric cancer according to comprehensive subgroup analyses, including age, sex, income, region of residence, obesity, and smoking (**a**), alcohol consumption, systolic blood pressure, diastolic blood pressure, fasting blood glucose, total cholesterol, CCI scores, and dyslipidemia history (**b**). The reference period is <180 days. Full results of the crude and adjusted overall weighted models are available in Appendix A (hydrophilic statins).

**Figure 3 pharmaceuticals-14-01283-f003:**
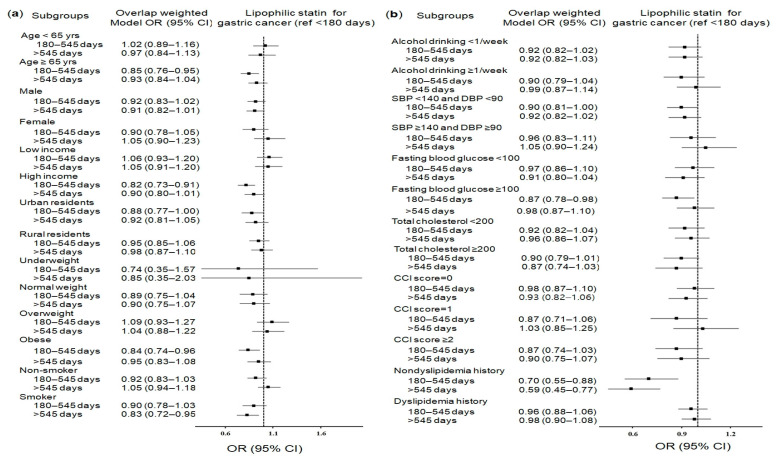
Forest plots for multivariable conditional logistic regression depicting the overlap weighted odds ratios (95% confidence intervals) of previous use duration of lipophilic statins for incident gastric cancer according to comprehensive subgroup analyses, including age, sex, income, region of residence, obesity, and smoking (**a**), alcohol consumption, systolic blood pressure, diastolic blood pressure, fasting blood glucose, total cholesterol, CCI scores, and dyslipidemia history (**b**). The reference period is <180 days. Full results of the crude and adjusted overall weighted models are available in Appendix A (lipophilic statins).

**Figure 4 pharmaceuticals-14-01283-f004:**
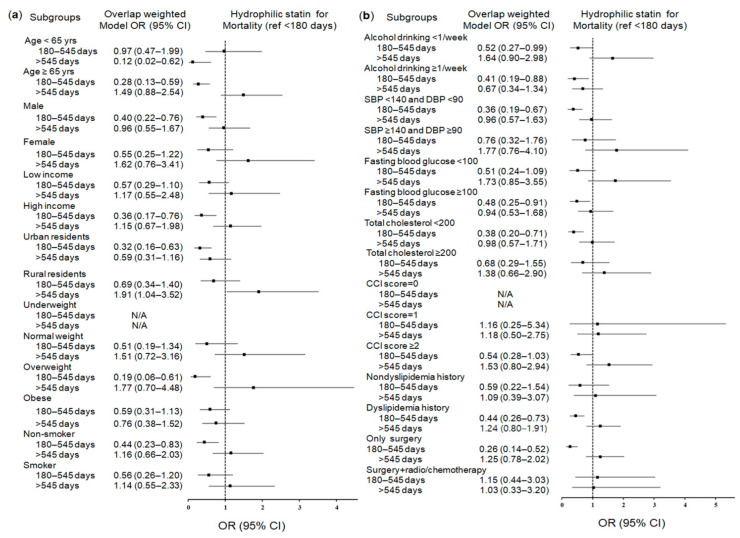
Forest plots for multivariable conditional logistic regression depicting the overlap weighted odds ratios (95% confidence intervals) of previous use duration of hydrophilic statins for overall mortality in the patients with incident gastric cancer according to comprehensive subgroup analyses, including age, sex, income, region of residence, obesity, and smoking (**a**), alcohol consumption, systolic blood pressure, diastolic blood pressure, fasting blood glucose, total cholesterol, CCI scores, and dyslipidemia history (**b**). The reference period is <180 days. Full results of the crude and adjusted overall weighted models are available in Appendix A (hydrophilic statins).

**Figure 5 pharmaceuticals-14-01283-f005:**
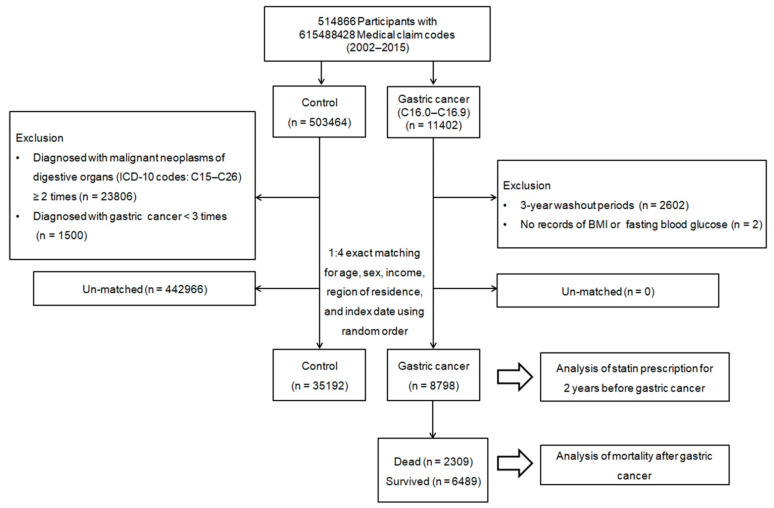
Flow illustration of participant selection. Of a total of 514,866 participants, 8798 gastric cancer patients were matched with 35,192 control participants for age, sex, income, and region of residence.

**Table 1 pharmaceuticals-14-01283-t001:** Baseline characteristics of participants.

Characteristics	Before Overlap Weighting Adjustment				After Overlap Weighting Adjustment	
	Gastric Cancer	Control	Standardized Difference	Gastric Cancer	Control	Standardized Difference
Age (%)			0.00			0.00
40–44	0.51	0.51		0.47	0.47	
45–49	4.54	4.54		4.27	4.27	
50–54	11.51	11.51		11.43	11.43	
55–59	13.46	13.46		13.47	13.47	
60–64	17.22	17.22		17.20	17.20	
65–69	17.47	17.47		17.46	17.46	
70–74	17.63	17.63		17.71	17.71	
75–79	10.9	10.9		11.03	11.03	
80–84	5.46	5.46		5.64	5.64	
85+	1.31	1.31		1.32	1.32	
Sex (%)			0.00			0.00
Male	73.55	73.55		73.18	73.17	
Female	26.45	26.45		26.82	26.83	
Income (%)			0.00			0.00
1 (lowest)	15.69	15.69		15.60	15.60	
2	12.33	12.33		12.32	12.32	
3	16.03	16.03		15.80	15.80	
4	21.46	21.46		21.64	21.64	
5 (highest)	34.5	34.5		34.64	34.64	
Region of residence (%)			0.00			0.00
Urban	41.08	41.08		40.99	40.99	
Rural	58.92	58.92		59.01	59.01	
Obesity ^†^ (%)			0.07			0.00
Underweight	3.78	2.76		3.54	3.54	
Normal	36.83	35.41		36.44	36.44	
Overweight	25.97	27.57		26.35	26.35	
Obese I	34.05	31.50		31.16	31.16	
Obese II	2.36	2.77		2.51	2.51	
Smoking status (%)			0.09			0.00
Nonsmoker	56.8	61.1		57.57	57.57	
Past smoker	19.52	17.58		19.48	19.48	
Current smoker	23.69	21.33		22.95	22.95	
Alcohol consumption (%)			0.09			0.00
<1 time a week	55.61	60.22		56.09	56.09	
≥1 time a week	44.39	39.78		43.91	43.91	0.00
SBP (Mean, SD)	128.43 (16.58)	128.58 (16.93)	0.01	128.50 (13.85)	128.51 (7.21)	0.00
DBP (Mean, SD)	78.74 (10.60)	78.96 (10.60)	0.02	78.78 (8.85)	78.78 (4.48)	0.00
Total cholesterol (Mean, SD)	193.15 (39.30)	195.84 (38.06)	0.07	193.90 (32.67)	193.90 (16.01)	0.00
Fasting blood glucose (Mean, SD)	104.32 (32.77)	102.73 (30.42)	0.05	103.98 (26.75)	103.98 (13.44)	0.00
CCI score (Mean, SD)	2.19 (2.63)	0.71 (1.37)	0.71	1.42 (1.68)	1.42 (0.87)	0.00
Dyslipidemia history (%)	37.33	42.07	0.10	39.51	39.51	0.00

Abbreviations: CCI, Charlson Comorbidity Index; SBP, systolic blood pressure; SD, standard deviation; DBP, diastolic blood pressure. ^†^ Obesity (BMI, body mass index, kg/m^2^) was categorized as <18.5 (underweight), ≥18.5 to <23 (normal), ≥23 to <25 (overweight), ≥25 to <30 (obese I), and ≥30 (obese II).

**Table 2 pharmaceuticals-14-01283-t002:** Crude and overlap propensity score weighted odds ratios of dates of statin prescription for gastric cancer according to statin type.

Characteristics	Gastric Cancer	Control	Odds Ratios for Gastric Cancer (95% Confidence Interval)			
	%	%	Crude	*p*-Value	Overlap Weighted Model ^†^	*p*-Value
Any statin						
<180 days	87.7	86.8	1		1	
180–545 days	5.7	6.3	0.89 (0.80–0.98)	0.021 *	0.88 (0.81–0.86)	0.002 *
>545 days	6.6	6.9	0.94 (0.86–1.04)	0.214	0.96 (0.89–1.04)	0.292
Hydrophilic statins						
<180 days	97.8	97.7	1		1	
180–545 days	1.1	1.3	0.85 (0.68–0.95)	0.149	0.78 (0.66–0.92)	0.004 *
>545 days	1.1	1.0	1.11 (0.89–1.39)	0.368	1.03 (0.86–1.23)	0.733
Lipophilic statins						
<180 days	89.7	88.8	1		1	
180–545 days	5.3	5.7	0.91 (0.82–1.01)	0.074	0.91 (0.84–0.99)	0.039 *
>545 days	5.0	5.5	0.91 (0.82–1.01)	0.085	0.96 (0.88–1.04)	0.327

The total numbers of gastric cancer patients and controls were 8789 and 35,192, respectively. * Significance at *p* < 0.05. ^†^ Adjusted for age, sex, income, region of residence, systolic blood pressure, diastolic blood pressure, fasting blood glucose, total cholesterol, obesity, smoking, alcohol consumption, dyslipidemia history, and Charlson Comorbidity Index (CCI) scores.

**Table 3 pharmaceuticals-14-01283-t003:** Baseline characteristics of gastric cancer participants.

Characteristics	Before Overlap Weighting Adjustment			After Overlap Weighting Adjustment		
	Deceased pts	Survived pts	Standardized Difference	Deceased pts	Survived pts	Standardized Difference
Age (%)			0.56			0.00
40–44	0.52	0.51		0.47	0.47	
45–49	3.94	4.75		3.82	3.82	
50–54	7.19	13.05		7.36	7.36	
55–59	8.32	15.29		9.00	9.00	
60–64	11.95	19.09		13.05	13.05	
65–69	16.46	17.83		17.52	17.52	
70–74	19.84	16.84		20.68	20.68	
75–79	17.63	8.51		16.75	16.75	
80–84	11.04	3.47		9.24	9.24	
85+	3.12	0.66		2.11	2.11	
Sex (%)			0.08			0.00
Male	76.27	72.58		74.75	74.75	
Female	23.73	27.42		25.25	25.25	
Income (%)			0.14			0.00
1 (lowest)	19.53	14.32		18.50	18.50	
2	12.69	12.21		13.00	13.0	
3	17.58	15.47		16.87	16.87	
4	19.45	22.18		20.74	20.74	
5 (highest)	30.75	35.83		30.89	30.89	
Residence (%)			0.08			0.00
Urban	38.03	42.16		38.31	38.31	
Rural	61.97	57.84		61.69	61.69	
Obesity ^†^ (%)			0.22			0.00
Underweight	7.28	2.54		6.12	6.12	
Normal	40.45	35.54		40.98	40.98	
Overweight	24.25	26.58		24.22	24.22	
Obese I	26.5	32.67		26.86	26.87	
Obese II	1.52	2.67		1.82	1.82	
Smoking (%)			0.05			0.00
Non	58.68	56.13		58.33	58.32	
Past	14.85	21.17		17.39	17.39	
Current	26.46	22.7		24.28	24.28	
Alcohol (%)			0.27			0.00
<1 time/week	65.31	52.17		61.99	61.98	
≥1 time/week	34.69	47.83		38.01	38.02	0.00
SBP (Mean, SD)	130.35 (18.07)	127.75 (15.96)	0.15	129.85 (11.75)	129.85 (6.57)	0.00
DBP (Mean, SD)	79.27 (11.28)	78.55 (10.34)	0.07	78.95 (7.29)	78.95 (4.08)	0.00
TCL (Mean, SD)	190.08 (41.15)	194.25 (38.57)	0.10	190.50 (26.82)	190.50 (15.06)	0.00
FBGL (Mean, SD)	106.25 (37.71)	103.64 (30.80)	0.07	106.29 (24.08)	106.29 (15.77)	0.00
CCI score (Mean, SD)	4.60 (2.82)	1.34 (1.94)	1.35	3.17 (1.74)	3.17 (1.09)	0.00
Dyslipidemia (%)	21.7	42.89	0.47	28.42	28.42	0.00
Treatment (%)			0.77			0.00
Only surgery	59.64	90.65		74.58	74.58	
Surgery + RT/CT	40.36	9.35		25.42	25.42	

Abbreviations: pts, patients; CCI, Charlson Comorbidity Index; SBP, systolic blood pressure; SD, standard deviation; DBP, diastolic blood pressure; TCL, total cholesterol level; FBGL, fasting blood glucose; RT, radiotherapy; CT, chemotherapy. ^†^ Obesity (BMI, body mass index, kg/m^2^) was categorized as <18.5 (underweight), ≥18.5 to <23 (normal), ≥23 to <25 (overweight), ≥25 to <30 (obese I), and ≥30 (obese II).

**Table 4 pharmaceuticals-14-01283-t004:** Crude and overlap propensity score weighted odds ratios of dates of statin prescriptions for overall mortality in gastric cancer participants.

Characteristics	Deceased	Survived	Odds Ratios for Mortality (95% Confidence Interval)			
	%	%	Crude	*p*-Value	Overlap Weighted Model ^†^	*p*-Value
Any statin						
<180 days	91.1	86.5	1		1	
180–545 days	4.7	6.0	0.74 (0.60–0.93)	0.008 *	1.15 (0.94–1.40)	0.164
>545 days	4.2	7.4	0.53 (0.42–0.67)	<0.001 *	0.85 (0.69–1.04)	0.106
Hydrophilic statins						
<180 days	98.5	97.5	1		1	
180–545 days	0.6	1.3	0.47 (0.27–0.83)	0.009 *	0.47 (0.29–0.77)	0.003 *
>545 days	0.9	1.2	0.75 (0.46–1.22)	0.243	1.17 (0.75–1.81)	0.495
Lipophilic statins						
<180 days	92.5	88.7	1		1	
180–545 days	4.4	5.6	0.76 (0.61–0.95)	0.016 *	1.21 (0.99–1.49)	0.064
>545 days	3.0	5.7	0.51 (0.39–0.66)	<0.001 *	0.82 (0.65–1.03)	0.086

The total numbers of deceased participants and surviving participants were 2309 and 6489, respectively. * Significance at *p* < 0.05. ^†^ Adjusted for age, sex, income, region of residence, systolic blood pressure, diastolic blood pressure, fasting blood glucose, total cholesterol, obesity, smoking, alcohol consumption, dyslipidemia history, and Charlson Comorbidity Index (CCI) scores.

## Data Availability

All data are available from the database of the National Health Insurance Sharing Service (NHISS) (https://nhiss.nhis.or.kr/NHISS) (3 January 2020), which allows any researcher full access to the data. The data used in this article can be downloaded from the website after promising to follow the research ethics.

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
