# Peer review of "Association between Statin Use and Gastric Cancer: A Nested Case-Control Study Using a National Health Screening Cohort in Korea"

_pharmaceuticals, 2021, doi:10.3390/ph14121283_

Round 1

Reviewer 1 Report

The paper by Kwon et al. analyzed the association between statin use and Gastric cancer in the Korean population. The introduction provides a good background of the topic. The objective of the paper is clearly defined. 

It seems the materials and methods section was originally placed after Introduction and moved to the end at the last minute. Due to this transfer, the rest of the text has not been carefully revised in terms of figure numbers, abbreviations, etc.

 My specific comments are listed below:

  • line 28: "and the duration of This nested..." - duration of what? I think something is missing here
  • line 43-45: sentence unclear, please consider rephrasing
  • introduction sentences 2 and 3: repeated information
  • line 102: CCI - abbreviation explained later in section 4
  • table 1: the table would be easier to read if it was placed on one page 
  • table 2: is the total number of N is really needed in every row of the table? please consider providing a total number in the text or table description. Please consider listing a number of N or percentage in a table, but not both. This will reduce the amount of information which may improve the readability of the table
  • page 6 - this should be figure 1 not 2, too much information, completely unreadable
  • table 3: see table 1
  • table 4: see table 2
  • page 10: figure 2 not 3, too much information, completely unreadable
  • line 231-235: sentence unclear
  • line 242-245: sentence unclear
  • line 312: abbreviation explained later in section 4
  • line 417, 419: please describe in brief

Author Response

Reviewer #1:

General Comments: The paper by Kwon et al. analyzed the association between statin use and gastric cancer in the Korean population. The introduction provides a good background of the topic. The objective of the paper is clearly defined.

It seems the materials and methods section was originally placed after Introduction and moved to the end at the last minute. Due to this transfer, the rest of the text has not been carefully revised in terms of figure numbers, abbreviations, etc.

Response: We thank you for the critical comments and helpful suggestions. We apologize for the incomplete final arrangement check for journal’s different arrangement and the corresponding abbreviation and the explanation, and Figure arrangements when submitted and inconvenience they caused in your reading. We have taken all these comments and suggestions into account and have made major corrections in this revised manuscript.

  1. Comment: line 28: "and the duration of This nested..." - duration of what? I think something is missing here

Response: We apologize for the error in the transferred PDF version from the MS word. The original MS word version showed missing words, but we checked the missing words in the PDF version, which may be due to the insertion of line numbering in the PDF version. We corrected the missing words in the new PDF version, as described below.

(Abstract: Page 1, line 28): Here, we investigated the likelihood of incident gastric cancers and the related mortality depending on statin exposure, statin type, and the duration of use. This nested case–control study included 8,798 patients who were diagnosed with gastric cancer and matched with 35,192 controls at a 1:4 ratio using propensity score matching from a nationwide database in Korea (2002–2015).

  1. 2. Comment: line 43-45: sentence unclear, please consider rephrasing.

Response: Thank you for your suggestion. We apologize for the poor English editing. We revised the sentences to try to express our clinical importance of the study, as described below.

(Abstract: Page 1, lines 44): Thus, the current nationwide cohort study suggests that prior short-term statin use may have anticancer benefits in elderly patients with hyperglycemia against gastric cancer.

  1. 3. Comment: introduction sentences 2 and 3: repeated information

Response: Thank you for your comments. We changed the sentences, as described below.

(Introduction: Page 2, line 55): Over the last two decades, it is clear that mandatory national endoscopic screening programs for gastric cancer in adults over 40 years of age every two years have reduced mortality from gastric cancer [3].

  1. 4. Comment: line 102: CCI - abbreviation explained later in section 4

Response: We apologize for inconvenience and thank you for your patience. We revised the abbreviation in order in the manuscript, as described below.

(Results: Page 3, line 114): Charlson Comorbidity Index (CCI) score

(Results: Page 3, line 123): odds ratio (OR)

(Results: Page 3, line 124): 95% confidence interval (CI)

(Results: Page 3, line 138): diastolic blood pressure (DBP)

(Results: Page 3, line 138): systolic blood pressure (SBP)

(Discussion: Page 12, line 298): hazard ratio

(Discussion; page 12, line 324): Korean National Health Insurance Service-Health Screening (KNHIS-HS)

(Discussion; page 12, line 345): Helicobacter pylori

  1. 5. Comment: table 1: the table would be easier to read if it was placed on one page

Response: Thank you for your suggestion. In fact, in the original manuscript in null form of MS Word, the Table 1 was within a single page, but after putting in the Pharmaceuticals Template, the Table size could not be changeable. I tried again to make smaller Table 1 within one page, which seems to be over my ability now. I will ask Editor to change Table 1 as fit as one page later, I promise.

Table 1. General Characteristics of Participants

Characteristics

Before Overlap weighting adjustment

After Overlap weighting adjustment

Gastric cancer

Control

Standardized

Difference

Gastric cancer

Control

Standardized

Difference

Age (%)

0.00

0.00

40-44

0.51

0.51

0.47

0.47

45-49

4.54

4.54

4.27

4.27

50-54

11.51

11.51

11.43

11.43

55-59

13.46

13.46

13.47

13.47

60-64

17.22

17.22

17.20

17.20

65-69

17.47

17.47

17.46

17.46

70-74

17.63

17.63

17.71

17.71

75-79

10.9

10.9

11.03

11.03

80-84

5.46

5.46

5.64

5.64

85+

1.31

1.31

1.32

1.32

Sex (%)

0.00

0.00

Male

73.55

73.55

73.18

73.17

Female

26.45

26.45

26.82

26.83

Income (%)

0.00

0.00

1 (lowest)

15.69

15.69

15.60

15.60

2

12.33

12.33

12.32

12.32

3

16.03

16.03

15.80

15.80

4

21.46

21.46

21.64

21.64

5 (highest)

34.5

34.5

34.64

34.64

Region of residence (%)

0.00

0.00

Urban

41.08

41.08

40.99

40.99

Rural

58.92

58.92

59.01

59.01

Obesity † (%)

0.07

0.00

Underweight

3.78

2.76

3.54

3.54

Normal

36.83

35.41

36.44

36.44

Overweight

25.97

27.57

26.35

26.35

Obese I

34.05

31.50

31.16

31.16

Obese II

2.36

2.77

2.51

2.51

Smoking status (%)

0.09

0.00

Nonsmoker

56.8

61.1

57.57

57.57

Past smoker

19.52

17.58

19.48

19.48

Current smoker

23.69

21.33

22.95

22.95

Alcohol consumption (%)

0.09

0.00

<1 time a week

55.61

60.22

56.09

56.09

≥1 time a week

44.39

39.78

43.91

43.91

0.00

SBP (Mean, SD)

128.43 (16.58)

128.58 (16.93)

0.01

128.50 (13.85)

128.51 (7.21)

0.00

DBP (Mean, SD)

78.74 (10.60)

78.96 (10.60)

0.02

78.78 (8.85)

78.78 (4.48)

0.00

Total cholesterol (Mean, SD)

193.15 (39.30)

195.84 (38.06)

0.07

193.90 (32.67)

193.90 (16.01)

0.00

Fasting blood glucose (Mean, SD)

104.32 (32.77)

102.73 (30.42)

0.05

103.98 (26.75)

103.98 (13.44)

0.00

CCI score (Mean, SD)

2.19 (2.63)

0.71 (1.37)

0.71

1.42 (1.68)

1.42 (0.87)

0.00

Dyslipidemia history (%)

37.33

42.07

0.10

39.51

39.51

0.00

Abbreviations: CCI, Charlson Comorbidity Index; SBP, systolic blood pressure; SD, standard deviation; DBP, diastolic blood pressure

† Obesity (BMI, body mass index, kg/m2) was categorized as < 18.5 (underweight), ≥ 18.5 to < 23 (normal), ≥ 23 to < 25 (overweight), ≥ 25 to < 30 (obese I), and ≥ 30 (obese II).

  1. 6. Comment: table 2: is the total number of N is truly needed in every row of the table? please consider providing a total number in the text or table description. Please consider listing a number of N or percentage in a table, but not both. This will reduce the amount of information, which may improve the readability of the table.

Response: Thank you for your suggestion. I agree with your opinion. As per your suggestions, we revised Table 2 as follows.

(Page 4; Table 2): Table 2. Crude and overlap propensity score weighted odds ratios of dates of statin prescription for gastric cancer according to statin type

Characteristics

Gastric cancer

Control

Odd ratios for gastric cancer (95% confidence interval)

%

%

Crude

P value

Overlap weighted model†

P value

Any statin

< 180 days

87.7

86.8

1

1

180 to 545 days

5.7

6.3

0.89 (0.80-0.98)

0.021*

0.88 (0.81-0.86)

0.002*

> 545 days

6.6

6.9

0.94 (0.86-1.04)

0.214

0.96 (0.89-1.04)

0.292

Hydrophilic statin

< 180 days

97.8

97.7

1

1

180 to 545 days

1.1

1.3

0.85 (0.68-0.95)

0.149

0.78 (0.66-0.92)

0.004*

> 545 days

1.1

1.0

1.11 (0.89-1.39)

0.368

1.03 (0.86-1.23)

0.733

Lipophilic statin

< 180 days

89.7

88.8

1

1

180 to 545 days

5.3

5.7

0.91 (0.82-1.01)

0.074

0.91 (0.84-0.99)

0.039*

> 545 days

5.0

5.5

0.91 (0.82-1.01)

0.085

0.96 (0.88-1.04)

0.327

The total numbers of gastric cancer patients and controls were 8,789 and 35,192, respectively.

*Significance at P < 0.05

†Adjusted for age, sex, income, region of regidence, systolic blood pressure, diastolic blood

pressure, fasting blood glucose, total cholesterol, obesity, smoking, alcohol consumption,

dyslipidemia history, and Charlson Comorbidity Index (CCI) scores.

  1. 7. Comment: page 6 - this should be figure 1 not 2, too much information, completely unreadable

Response: We apologize for inconvenience due to poor Figure editing and thank you for your patience.

We agree with your opinion. We tried to make separate Figures, as like below. The figure arrangements will be improved by a professional graphic manager of the journal.

(Result: Figures 1-3):

Figure 1. Forest plots for multivariable conditional logistic regression depicting the overlap weighted odds ratios (95% confidence intervals) of previous use duration of any statin for incident gastric cancer according to comprehensive subgroup analyses including age, sex, income, region of residence, obesity, and smoking (a), alcohol consumption, systolic blood pressure, diastolic blood pressure, fasting blood glucose, total cholesterol, CCI scores, and dyslipidemia history (b). The reference period is <180 days. Full results of the crude and adjusted overall weighted models are available in Supplementary Table S1 (any statin).

Figure 2. Forest plots for multivariable conditional logistic regression depicting the overlap weighted odds ratios (95% confidence intervals) of previous use duration of hydrophilic statin for incident gastric cancer according to comprehensive subgroup analyses including age, sex, income, region of residence, obesity, and smoking (a), alcohol consumption, systolic blood pressure, diastolic blood pressure, fasting blood glucose, total cholesterol, CCI scores, and dyslipidemia history (b). The reference period is <180 days. Full results of the crude and adjusted overall weighted models are available in Supplementary Table S2 (hydrophilic statin).

Figure 3. Forest plots for multivariable conditional logistic regression depicting the overlap weighted odds ratios (95% confidence intervals) of previous use duration of lipophilic statin for incident gastric cancer according to comprehensive subgroup analyses including age, sex, income, region of residence, obesity, and smoking (a), alcohol consumption, systolic blood pressure, diastolic blood pressure, fasting blood glucose, total cholesterol, CCI scores, and dyslipidemia history (b). The reference period is <180 days. Full results of the crude and adjusted overall weighted models are available in Supplementary Table S3 (lipophilic statin).

  1. Comment: table 3: see table 1 (table would be easier to read if it was placed on one page)

Response: Thank you for your suggestion, and I have to deeply say sorry to you for the limitation of my ability to edit MS words in the fixed form of journal Template. Once again I have to excuse the fact, in the original manuscript in null form of MS Word, the Table 3 was within a single page, but after putting in the Pharmaceuticals Template, the Table size could not be changeable. I tried again to make smaller Table 3 within one page, which seems to be over my ability now. I will ask professional Editor to change Table 3 as fit as one page later. Thank you for your patience.

Table 3. Baseline characteristics of gastric cancer participants.

Characteristics

Before Overlap weighting adjustment

After Overlap weighting adjustment

Deceased pts

Survived pts

Standardized

Difference

Deceased pts

Survived pts

Standardized

Difference

Age (%)

0.56

0.00

40-44

0.52

0.51

0.47

0.47

45-49

3.94

4.75

3.82

3.82

50-54

7.19

13.05

7.36

7.36

55-59

8.32

15.29

9.00

9.00

60-64

11.95

19.09

13.05

13.05

65-69

16.46

17.83

17.52

17.52

70-74

19.84

16.84

20.68

20.68

75-79

17.63

8.51

16.75

16.75

80-84

11.04

3.47

9.24

9.24

85+

3.12

0.66

2.11

2.11

Sex (%)

0.08

0.00

Male

76.27

72.58

74.75

74.75

Female

23.73

27.42

25.25

25.25

Income (%)

0.14

0.00

1 (lowest)

19.53

14.32

18.50

18.50

2

12.69

12.21

13.00

13.0

3

17.58

15.47

16.87

16.87

4

19.45

22.18

20.74

20.74

5 (highest)

30.75

35.83

30.89

30.89

Residence (%)

0.08

0.00

Urban

38.03

42.16

38.31

38.31

Rural

61.97

57.84

61.69

61.69

Obesity † (%)

0.22

0.00

Underweight

7.28

2.54

6.12

6.12

Normal

40.45

35.54

40.98

40.98

Overweight

24.25

26.58

24.22

24.22

Obese I

26.5

32.67

26.86

26.87

Obese II

1.52

2.67

1.82

1.82

Smoking (%)

0.05

0.00

Non

58.68

56.13

58.33

58.32

Past

14.85

21.17

17.39

17.39

Current

26.46

22.7

24.28

24.28

Alcohol (%)

0.27

0.00

<1 time/week

65.31

52.17

61.99

61.98

≥1 time/week

34.69

47.83

38.01

38.02

0.00

SBP (Mean, SD)

130.35 (18.07)

127.75 (15.96)

0.15

129.85 (11.75)

129.85 (6.57)

0.00

DBP (Mean, SD)

79.27 (11.28)

78.55 (10.34)

0.07

78.95 (7.29)

78.95 (4.08)

0.00

TCL (Mean, SD)

190.08 (41.15)

194.25 (38.57)

0.10

190.50 (26.82)

190.50 (15.06)

0.00

FBGL (Mean, SD)

106.25 (37.71)

103.64 (30.80)

0.07

106.29 (24.08)

106.29 (15.77)

0.00

CCI score (Mean, SD)

4.60 (2.82)

1.34 (1.94)

1.35

3.17 (1.74)

3.17 (1.09)

0.00

Dyslipidemia (%)

21.7

42.89

0.47

28.42

28.42

0.00

Treatment (%)

0.77

0.00

Only surgery

59.64

90.65

74.58

74.58

Surgery+RT/CT

40.36

9.35

25.42

25.42

Abbreviations: pts, patients; CCI, Charlson Comorbidity Index; SBP, Systolic blood pressure; SD, standard deviation; DBP, Diastolic blood pressure; TCL, Total cholesterol level; FBGL, Fasting blood glucose; RT, Radiotherapy; CT, Chemotherapy.

† Obesity (BMI, body mass index, kg/m2) was categorized as < 18.5 (underweight), ≥ 18.5 to < 23 (normal), ≥ 23 to < 25 (overweight), ≥ 25 to < 30 (obese I), and ≥ 30 (obese II).

  1. Comment: table 4: see table 2 (is the total number of N is truly needed in every row of the table? please consider providing a total number in the text or table description. Please consider listing a number of N or percentage in a table, but not both. This will reduce the amount of information, which may improve the readability of the table)

Response: Thank you for your suggestion. I agree with your opinion. As per your suggestions, we revised Table 4 as follows.

Table 4. Crude and overlap propensity score weighted odds ratios of dates of statin prescription for overall mortality in gastric cancer participants.

Characteristics

Deceased

Survived

Odd ratios for mortality (95% confidence interval)

%

%

Crude

P value

Overlap weighted model†

P value

Any statin

< 180 days

91.1

86.5

1

1

180 to 545 days

4.7

6.0

0.74 (0.60-0.93)

0.008*

1.15 (0.94-1.40)

0.164

> 545 days

4.2

7.4

0.53 (0.42-0.67)

<0.001*

0.85 (0.69-1.04)

0.106

Hydrophilic statin

< 180 days

98.5

97.5

1

1

180 to 545 days

0.6

1.3

0.47 (0.27-0.83)

0.009*

0.47 (0.29-0.77)

0.003*

> 545 days

0.9

1.2

0.75 (0.46-1.22)

0.243

1.17 (0.75-1.81)

0.495

Lipophilic statin

< 180 days

92.5

88.7

1

1

180 to 545 days

4.4

5.6

0.76 (0.61-0.95)

0.016*

1.21 (0.99-1.49)

0.064

> 545 days

3.0

5.7

0.51 (0.39-0.66)

<0.001*

0.82 (0.65-1.03)

0.086

The total numbers of deceased participants and surviving participants were 2,309 and 6,489, respectively.

* Significance at P < 0.05

† Adjusted for age, sex, income, region of regidence, systolic blood pressure, diastolic blood pressure, fasting blood glucose, total cholesterol, obesity, smoking, alcohol consumption, dyslipidemia history, and Charlson Comorbidity Index (CCI) scores.

  1. Comment: page 10: figure 2 not 3, too much information, completely unreadable

Response: We apologize for inconvenience due to poor Figure editing and thank you for your patience.

We agree with your opinion. We tried to make separate Figures, as like below. We retained Figure 4 about the effect of hydrophilic statins on mortality in the main manuscript. Other figures regarding any statin and lipophilic statin on mortality were moved to Supplementary Figures 1 and 2. The figure arrangements are preliminary now and will be improved by professional graphic managers of the journal.

(Figure 4)

Figure 4. Forest plots for multivariable conditional logistic regression depicting the overlap weighted odds ratios (95% confidence intervals) of previous use duration of hydrophilic statin for overall mortality in the patients with incident gastric cancer according to comprehensive subgroup analyses including age, sex, income, region of residence, obesity, and smoking (a), alcohol consumption, systolic blood pressure, diastolic blood pressure, fasting blood glucose, total cholesterol, CCI scores, and dyslipidemia history (b). The reference period is <180 days. Full results of the crude and adjusted overall weighted models are available in Supplementary Tables S5 (hydrophilic statin).

Supplementary Figure 1. Forest plots for multivariable conditional logistic regression depicting the overlap weighted odds ratios (95% confidence intervals) of previous use duration of any statin for overall mortality in the patients with incident gastric cancer according to comprehensive subgroup analyses including age, sex, income, region of residence, obesity, and smoking (a), alcohol consumption, systolic blood pressure, diastolic blood pressure, fasting blood glucose, total cholesterol, CCI scores, and dyslipidemia history (b). The reference period is <180 days. Full results of the crude and adjusted overall weighted models are available in Supplementary Tables S4 (any statin).

Supplementary Figure 2. Forest plots for multivariable conditional logistic regression depicting the overlap weighted odds ratios (95% confidence intervals) of previous use duration of lipophilic statin for overall mortality in the patients with incident gastric cancer according to comprehensive subgroup analyses including age, sex, income, region of residence, obesity, and smoking (a), alcohol consumption, systolic blood pressure, diastolic blood pressure, fasting blood glucose, total cholesterol, CCI scores, and dyslipidemia history (b). The reference period is <180 days. Full results of the crude and adjusted overall weighted models are available in Supplementary Table S6 (lipophilic statin).

  1. Comment: line 231-235: sentence unclear

Response: We apologize for the grammar errors and poor English expression in this manuscript and inconvenience they caused in your reading. In fact, we edited the manuscript by a highly qualified native English speaking editor company. We revised the related sentences again, as described below.

(Discussion: Page 11, lines 255): The protective relevance of statins for mortality reduction in our study corresponds with the results from English and Scottish population-based studies. That study indicated a relationship between prior statin use and decreased mortality (hazard ratio=0.91; 95% CI=0.84–0.98) [30], which showed a much lower reduction in mortality (9%) than 53% reduced mortality in our study (OR=0.47; 95% CI=0.29–0.77).

  1. Comment: line 242-245: sentence unclear

Response: We apologize for the poor grammar in this manuscript and inconvenience they caused in your reading. In fact, we edited the manuscript by highly qualified native English speaking editors.

We revised the related sentences more clearly, as described below.

(Discussion: Page 11, line 265): A meta-analysis dealing with case–control studies also demonstrated no correlation between the cumulative use duration of statins and gastric cancer risk across global populations in Asia, Europe, or the USA [23].

  1. Comment: line 312: abbreviation explained later in section 4

Response: We apologize for the text arrangement errors in this manuscript and the inconvenience they caused in your reading. We have revised and checked all abbreviations in order and their corresponding full terms, as described below. I apologize for your inconvenience.

(Results: Page 3, line 114): Charlson Comorbidity Index (CCI) score

(Results: Page 3, line 123): odds ratio (OR)

(Results: Page 3, line 124): 95% confidence interval (CI)

(Results: Page 3, line 138): diastolic blood pressure (DBP)

(Results: Page 3, line 138): systolic blood pressure (SBP)

(Discussion: Page 12, line 298): hazard ratio

(Discussion; page 12, line 324): Korean National Health Insurance Service-Health Screening (KNHIS-HS)

(Discussion; page 12, line 345): Helicobacter pylori

  1. Comment: line 417, 419: please describe in brief

Response: Thank you for your suggestion. We tried to explain them in more detail, as described below.

(Material and Methods: Page 15, line 442):

Briefly, the participants were initially divided by income into 41 classes (one health aid class, 20 self-employed health insurance classes, and 20 employed health insurance classes). These groups were then recategorized into five classes (class 1 [lowest income]–class 5 [highest income]) [36]. Region of residence was divided into 16 areas according to administrative district. These regions were stratified based on urban (Seoul, Busan, Daegu, Incheon, Gwangju, Daejeon, and Ulsan) and rural (Gyeonggi, Gangwon, Chungcheongbuk, Chungcheongnam, Jeollabuk, Jeollanam, Gyeongsangbuk, Gyeongsangnam, and Jeju) areas [36]. Obesity status using BMI (kg/m2; <18.5 for underweight, ≥18.5 to <23 for normal weight, ≥23 to <25 for overweight, ≥25 to <30 for obese I, ≥30 for obese II), alcohol consumption (<1 time a week, ≥1 time a week), and smoking status (nonsmoker, past smoker, current smoker) were categorized in a similar manner as our previous study [37].

Reviewer 2 Report

Manuscript presented by Kwon and co-workers entitled “Association between Statin Use and Gastric Cancer: A Nested Case-Control Study Using a National Health Screening Cohort in Korea” is a research article presenting a huge study of the  likelihood of incident gastric cancers and the related mortality depending on statin exposure and type. The presented manuscript, before publication in Pharmaceuticals, needs major revision. Text is not properly organized, figures are not correctly presented. However, discussion is based on the lots of data, is not correctly presented.

My major comments are presented below. Provide the explanation for all of them, make changes in the text.

Major concerns:

- Abstract – lines 27-28 - Here, we investigated the likelihood of incident gastric  cancers and the related mortality depending on statin exposure, statin type, and the duration of – please finish the sentence. Duration of …?

- Abstract – present the novelty, utility and importance of the presented work

- Introduction – other risk factors should be briefly presented

- Introduction – describe briefly methods used in the analysis of statin as risk factor of cancer

- Introduction – briefly describe the statin types and their role in the cancer

- Introduction - Present the novelty of the article

- Introduction – in vitro should be in italics

- Figure 2 presented at page 7 (it should be figure 1) is of low quality and it is difficult to find some data in it

- Figure 3 (should be figure 2) is of low quality and it is difficult to find some data in it

- text should be justified

- discussion - discussion is based on lots of data however, is not correctly presented.

- the organization of  the text is not so good because at the page 15 Figure 1 is presented (of low quality)

- tables presented in the appendix are also presented in the supplementary data. In my opinion it will be better to present them in supplementary data only.

- In 12 of the papers published by the Hyo Geun Choi group there is a title in the form of “A Nested Case-Control Study” based on the ORCID data. Why is it important?

- unify the reference style

- check and correct English

Author Response

Reviewer #2:

General Comments: Manuscript presented by Kwon and coworkers entitled “Association between Statin Use and Gastric Cancer: A Nested Case–Control Study Using a National Health Screening Cohort in Korea” is a research article presenting a huge study of the likelihood of incident gastric cancers and the related mortality depending on statin exposure and type. The presented manuscript, before publication in Pharmaceuticals, needs major revision. Text is not properly organized, figures are not correctly presented. However, discussion is based on the lots of data, is not correctly presented.

My major comments are presented below. Provide the explanation for all of them, make changes in the text.

Response: We thank you for the critical comments and helpful suggestions. We have taken all these comments and suggestions into account and have made major corrections in this revised manuscript.

  1. Comment: Abstract – lines 27-28 - Here, we investigated the likelihood of incident gastric  cancers and the related mortality depending on statin exposure, statin type, and the duration of – please finish the sentence. Duration of …?

Response: We apologize for the error in the transferred PDF version from the MS word. The original MS word version showed missing words, but we checked the missing words in the PDF version, which may be due to the insertion of line numbering in the PDF version. We corrected the missing words in the new PDF version, as described below.

(Abstract: Page 1, line 28): Here, we investigated the likelihood of incident gastric cancers and the related mortality depending on statin exposure, statin type, and the duration of use. This nested case–control study included 8,798 patients who were diagnosed with gastric cancer and matched with 35,192 controls at a 1:4 ratio using propensity score matching from a nationwide database in Korea (2002–2015).

  1. 2. Comment: Abstract – present the novelty, utility and importance of the presented work

Response: Thank you for your suggestion.

We think the novelty of the study is that the subgroup analyses may herald the beneficial effective patients from prior statin use against gastric cancer risk and mortality, especially in elderly patients over 65 years old, normal blood pressure, and hyperglycemia. We added the clinical importance of the current nationwide cohort study to the Results and Conclusion sections in the abstract, as described below.

(Abstract: Page 1, line 41): In subgroup analyses, beneficial effects on both cancer development and mortality persisted in patients ≥65 years old, patients with normal blood pressure, and patients with high fasting glucose levels. There were no such associations with long-term statin use (>545 days). Thus, the current nationwide cohort study suggests that prior short-term statin use may have anticancer benefits in elderly patients with hyperglycemia against gastric cancer.

  1. 3. Comment: Introduction – other risk factors should be briefly presented

Response: Thank you for your suggestion. We added an explanation for risk factors for gastric cancers and revised the contents, as described below.

(Introduction: Page 2, line 52): The important risk factors for gastric cancer include advanced age, male sex, family history, Helicobacter pylori, history of chronic atrophic gastritis or pernicious anemia, obesity, chemical carcinogen use, smoking, red meat, alcohol, and low socioeconomic status [2].

  1. 4. Comment: Introduction – describe briefly methods used in the analysis of statin as risk factor for cancer

Response: Thank you for your suggestion. Initially, the risk factor for cancer development regarding statins comes from clinical trials of the WOSCOPS population with a 15-year follow-up that reported a greater incidence of prostate cancer in patients originally treated with pravastatin. These reports raised concern for cancer risk factors such as statin use. This possible mechanism may be involved in regulatory T cells that impair the host native and acquired antitumor immune response. We added this explanation to the introduction, as you suggested, as described below.

(Introduction: Page 2, line 70): Prospective data from clinical trials suggest that statins actually increase the incidences of breast cancer and prostate cancer long after over 15 follow-up years of statin treatment, possibly due to a statin-induced increase in regulatory T cells, resulting in impaired host antitumor immunity [8-10].

  1. 5. Comment: Introduction – briefly describe the statin types and their role in the cancer

Response: Thank you for your suggestion. Statin types are classified into hydrophilic and lipophilic according to hydrophobicity properties. Because of the high hepatic washout selectivity of hydrophilic statins, lipophilic statins have been considered more effective in nonhepatic solid organs, including breast and ovarian cancers. We added these explanations to the introduction and revised the related sentences, as described below.

(Introduction: Page 2, line 74): Statin types by hydrophobicity properties (hydrophilic or lipophilic) may influence the antiproliferative effects in cancers. Because of the high hepatic washout selectivity of hydrophilic statins and high penetration ability into the cell plasma membrane of lipophilic statins, lipophilic statins may be seemingly effective in nonhepatic, nonintestinal, solid organ cancers, including breast and ovarian cancers [11-13].

  1. 6. Comment Introduction - Present the novelty of the article

Response: Thank you for your suggestion. We think our novelty of the article is to demonstrate prior short-term statin use would reduce the risk for gastric cancer and its mortality, especially in elderly patients with hyperglycemia using nationwide large cohort data, which is of great clinical significance in aging society. We added our novelty and revised the related sentences, as described below.

(Introduction:  Page 2, line 89): There are two nationwide epidemiological studies regarding the influence of prior statin use on the development of stomach cancers based on Korean individuals [16,24]. These studies did not specifically investigate the relationships of the duration of statin use or statin types with the incidence and mortality of gastric cancer, and whether prior statin use affects the risk or mortality of gastric cancer according to the statin types and duration of use remains debated.

(Introduction:  Page 2, line 99): This study extended previous studies in the field; we further investigated potential risk factors related to statin use that may be predictive of incident gastric cancer and its mortality according to the statin type and duration of use. To explore this, a nationwide cohort study with an exactly matched nested case–control design was conducted, together with comprehensive subgroup analyses, to estimate the potential impacts of statins on the incidence and mortality of subsequent gastric cancers.

(Discussion:  Page 10 , line 226): In this large nationwide cohort study, we demonstrated that prior short-term statin use reduced the likelihood of gastric cancer and its overall mortality regardless of statin type and hydrophilic statin. This anticancer effect of statins in both reduced risk for gastric cancer and mortality persisted in patients aged over 65, patients with normal blood pressure, and patients with high fasting glucose levels. Since gastric cancer accounts for one of the prevalent malignancies in elderly patients aged over 65 in Korea [1], the most noteworthy finding is that statin use has reduced the development and mortality of a subset of gastric cancers in elderly individuals suffering from hyperglycemia.

  1. 7. Comment: Introduction – in vitro should be in italics.

Response: Thank you for your correction. We corrected it into italics, as like below.

(Introduction: Page 2, line 70): In vitro evidence indicates the anticancer activity of statins in gastric cancers, primarily by means of downregulation of the mevalonate pathway

  1. Comment: Figure 2 presented at page 7 (it should be figure 1) is of low quality and it is difficult to find some data in it.

Response: We apologize for inconvenience due to poor Figure editing and thank you for your patience. We agree with your opinion. We tried to make separate Figures, as like below. The figure arrangements will be improved by a professional graphic manager of the journal.

(Result: Figures 1-3):

Figure 1. Forest plots for multivariable conditional logistic regression depicting the overlap weighted odds ratios (95% confidence intervals) of previous use duration of any statin for incident gastric cancer according to comprehensive subgroup analyses including age, sex, income, region of residence, obesity, and smoking (a), alcohol consumption, systolic blood pressure, diastolic blood pressure, fasting blood glucose, total cholesterol, CCI scores, and dyslipidemia history (b). The reference period is <180 days. Full results of the crude and adjusted overall weighted models are available in Supplementary Table S1 (any statin).

Figure 2. Forest plots for multivariable conditional logistic regression depicting the overlap weighted odds ratios (95% confidence intervals) of previous use duration of hydrophilic statin for incident gastric cancer according to comprehensive subgroup analyses including age, sex, income, region of residence, obesity, and smoking (a), alcohol consumption, systolic blood pressure, diastolic blood pressure, fasting blood glucose, total cholesterol, CCI scores, and dyslipidemia history (b). The reference period is <180 days. Full results of the crude and adjusted overall weighted models are available in Supplementary Table S2 (hydrophilic statin).

Figure 3. Forest plots for multivariable conditional logistic regression depicting the overlap weighted odds ratios (95% confidence intervals) of previous use duration of lipophilic statin for incident gastric cancer according to comprehensive subgroup analyses including age, sex, income, region of residence, obesity, and smoking (a), alcohol consumption, systolic blood pressure, diastolic blood pressure, fasting blood glucose, total cholesterol, CCI scores, and dyslipidemia history (b). The reference period is <180 days. Full results of the crude and adjusted overall weighted models are available in Supplementary Table S3 (lipophilic statin).

  1. Comment: Figure 3 (should be figure 2) is of low quality and it is difficult to find some data in it.

Response: We apologize for inconvenience due to poor Figure editing and thank you for your patience.

We agree with your opinion. We retained Figure 4 about the effect of hydrophilic statins on mortality in the main manuscript. Other figures regarding any statin and lipophilic statin on mortality were moved to Supplementary Figures 1 and 2. The figure arrangements are preliminary now and will be improved by professional graphic managers of the journal. We tried to make separate Figures, as like below.

(Figure 4)

Figure 4. Forest plots for multivariable conditional logistic regression depicting the overlap weighted odds ratios (95% confidence intervals) of previous use duration of hydrophilic statin for overall mortality in the patients with incident gastric cancer according to comprehensive subgroup analyses including age, sex, income, region of residence, obesity, and smoking (a), alcohol consumption, systolic blood pressure, diastolic blood pressure, fasting blood glucose, total cholesterol, CCI scores, and dyslipidemia history (b). The reference period is <180 days. Full results of the crude and adjusted overall weighted models are available in Supplementary Tables S5 (hydrophilic statin).

Supplementary Figure 1. Forest plots for multivariable conditional logistic regression depicting the overlap weighted odds ratios (95% confidence intervals) of previous use duration of any statin for overall mortality in the patients with incident gastric cancer according to comprehensive subgroup analyses including age, sex, income, region of residence, obesity, and smoking (a), alcohol consumption, systolic blood pressure, diastolic blood pressure, fasting blood glucose, total cholesterol, CCI scores, and dyslipidemia history (b). The reference period is <180 days. Full results of the crude and adjusted overall weighted models are available in Supplementary Tables S4 (any statin).

Supplementary Figure 2. Forest plots for multivariable conditional logistic regression depicting the overlap weighted odds ratios (95% confidence intervals) of previous use duration of lipophilic statin for overall mortality in the patients with incident gastric cancer according to comprehensive subgroup analyses including age, sex, income, region of residence, obesity, and smoking (a), alcohol consumption, systolic blood pressure, diastolic blood pressure, fasting blood glucose, total cholesterol, CCI scores, and dyslipidemia history (b). The reference period is <180 days. Full results of the crude and adjusted overall weighted models are available in Supplementary Table S6 (lipophilic statin).

  1. Comment: - text should be justified

Response: We apologize for the mistake of text arrangement in this manuscript according to the journal’s standard and the inconvenience they caused in your reading. We have corrected and checked again all abbreviations and their corresponding full terms, as described below. I apologize for your inconvenience.

(Results: Page 3, line 114): Charlson Comorbidity Index (CCI) score

(Results: Page 3, line 123): odds ratio (OR)

(Results: Page 3, line 124): 95% confidence interval (CI)

(Results: Page 3, line 138): diastolic blood pressure (DBP)

(Results: Page 3, line 138): systolic blood pressure (SBP)

(Discussion: Page 12, line 298): hazard ratio

(Discussion; page 12, line 324): Korean National Health Insurance Service-Health Screening (KNHIS-HS)

(Discussion; page 12, line 345): Helicobacter pylori

  1. Comment: - discussion - discussion is based on lots of data however, is not correctly presented.

Response: Thank you for your comments. We tried to revise the discussion more clearly and condensed it in context.

  1. Comment: - the organization of the text is not so good because at the page 15 Figure 1 is presented (of low quality)

Response: We deeply apologize for the poor text organization in this manuscript according to the journal’s standard and the inconvenience they caused in your reading. This journal principle is different from other journals. This journal put the Material and methods at the last part. We had to use a journal-specific template and had to put in person one by one. I agree with I was new and immature to this form of submission. I pledge to myself to be alert to the new formation. I am truly sorry. I apologize for your inconvenience.

The figures and their arrangements will be professionally edited by the journal’s graphic manager.

I revised the figure numbering, Figure 1, to Figure 5, as described below.

Figure 5. Flow illustration of participant selection. Of a total of 514,866 participants, 8,798 gastric cancer patients were matched with 35,192 control participants for age, sex, income, and region of residence.

  1. Comment: - tables presented in the appendix are also presented in the supplementary data. In my opinion it will be better to present them in supplementary data only.

Response: Thank you for your suggestion. All are my faults and my immaturity to journal policy. The supplementary tables in the Appendix were moved to separate Supplementary materials.

  1. Comment: - In 12 of the papers published by the Hyo Geun Choi group there is a title in the form of “A Nested Case–Control Study” based on the ORCID data. Why is it important?

Response: We marked "A nested case control study" because the basic study design could be explained by this.

From this, the reader could know what variable we chose first, what variables we could analyze, and what kinds of statistics used. Thus, from this, we can give vital information to readers. We explained in the Methods section the reason why we selected a nested case–control design, as described below.

(Material and methods: Page 13, line 365): Since a nested case–control study design is suitable to retrospectively identify causal associations of the history of subjects for the presence or absence of an exposure in outcome status at the outset of the investigation [34], we used a nested case–control design for the study.

  1. Comment: - unify the reference style

Response: Thank you for your comments. We originally used EndNote of “Pharmacuticals” to make references properly according to the journal style. We tried to check again the reference style, as described below.

(References):

References

  1. Jung, K.W.; Won, Y.J.; Hong, S.; Kong, H.J.; Im, J.S.; Seo, H.G. Prediction of Cancer Incidence and Mortality in Korea, 2021. Cancer Res. Treat. 2021, 53, 316-322, doi:10.4143/crt.2021.290.
  2. Hundahl, S.A.; Phillips, J.L.; Menck, H.R. The National Cancer Data Base Report on poor survival of U.S. gastric carcinoma patients treated with gastrectomy: Fifth Edition American Joint Committee on Cancer staging, proximal disease, and the "different disease" hypothesis. Cancer 2000, 88, 921-932.
  3. Jun, J.K.; Choi, K.S.; Lee, H.Y.; Suh, M.; Park, B.; Song, S.H.; Jung, K.W.; Lee, C.W.; Choi, I.J.; Park, E.C., et al. Effectiveness of the Korean National Cancer Screening Program in Reducing Gastric Cancer Mortality. Gastroenterology 2017, 152, 1319-1328 e1317, doi:10.1053/j.gastro.2017.01.029.
  4. Iannelli, F.; Lombardi, R.; Milone, M.R.; Pucci, B.; De Rienzo, S.; Budillon, A.; Bruzzese, F. Targeting Mevalonate Pathway in Cancer Treatment: Repurposing of Statins. Pat. Anticancer Drug Discov. 2018, 13, 184-200, doi:10.2174/1574892812666171129141211.
  5. Ortiz, N.; Delgado-Carazo, J.C.; Diaz, C. Importance of Mevalonate Pathway Lipids on the Growth and Survival of Primary and Metastatic Gastric Carcinoma Cells. Exp. Gastroenterol. 2021, 14, 217-228, doi:10.2147/CEG.S310235.
  6. Liu, Q.; Xia, H.; Zhou, S.; Tang, Q.; Zhou, J.; Ren, M.; Bi, F. Simvastatin Inhibits the Malignant Behaviors of Gastric Cancer Cells by Simultaneously Suppressing YAP and beta-Catenin Signaling. Targets Ther. 2020, 13, 2057-2066, doi:10.2147/OTT.S237693.
  7. Nielsen, S.F.; Nordestgaard, B.G.; Bojesen, S.E. Statin use and reduced cancer-related mortality. Engl. J. Med. 2012, 367, 1792-1802, doi:10.1056/NEJMoa1201735.
  8. Ford, I.; Murray, H.; Packard, C.J.; Shepherd, J.; Macfarlane, P.W.; Cobbe, S.M.; West of Scotland Coronary Prevention Study, G. Long-term follow-up of the West of Scotland Coronary Prevention Study. Engl. J. Med.2007, 357, 1477-1486, doi:10.1056/NEJMoa065994.
  9. Goldstein, M.R.; Mascitelli, L.; Pezzetta, F. Do statins prevent or promote cancer? Oncol. 2008, 15, 76-77, doi:10.3747/co.v15i2.235.
  10. Bates, G.J.; Fox, S.B.; Han, C.; Leek, R.D.; Garcia, J.F.; Harris, A.L.; Banham, A.H. Quantification of regulatory T cells enables the identification of high-risk breast cancer patients and those at risk of late relapse. Clin. Oncol. 2006, 24, 5373-5380, doi:10.1200/JCO.2006.05.9584.
  11. Vogel, T.J.; Goodman, M.T.; Li, A.J.; Jeon, C.Y. Statin treatment is associated with survival in a nationally representative population of elderly women with epithelial ovarian cancer. Oncol. 2017, 146, 340-345, doi:10.1016/j.ygyno.2017.05.009.
  12. Wu, X.D.; Zeng, K.; Xue, F.Q.; Chen, J.H.; Chen, Y.Q. Statins are associated with reduced risk of gastric cancer: a meta-analysis. J. Clin. Pharmacol. 2013, 69, 1855-1860, doi:10.1007/s00228-013-1547-z.
  13. Liu, B.; Yi, Z.; Guan, X.; Zeng, Y.X.; Ma, F. The relationship between statins and breast cancer prognosis varies by statin type and exposure time: a meta-analysis. Breast Cancer Res. Treat. 2017, 164, 1-11, doi:10.1007/s10549-017-4246-0.
  14. Yang, H.C.; Islam, M.M.; Nguyen, P.A.A.; Wang, C.H.; Poly, T.N.; Huang, C.W.; Li, Y.J. Development of a Web-Based System for Exploring Cancer Risk With Long-term Use of Drugs: Logistic Regression Approach. JMIR Public Health Surveill. 2021, 7, e21401, doi:10.2196/21401.
  15. Yang, P.R.; Tsai, Y.Y.; Chen, K.J.; Yang, Y.H.; Shih, W.T. Statin Use Improves Overall Survival of Patients with Gastric Cancer after Surgery and Adjuvant Chemotherapy in Taiwan: A Nationwide Matched Cohort Study. Cancers (Basel) 2020, 12, doi:10.3390/cancers12082055.
  16. You, H.S.; You, N.; Lee, J.W.; Lim, H.J.; Kim, J.; Kang, H.T. Inverse Association between Statin Use and Stomach Cancer Incidence in Individuals with Hypercholesterolemia, from the 2002-2015 NHIS-HEALS Data. J. Environ. Res. Public Health 2020, 17, doi:10.3390/ijerph17031054.
  17. Lee, J.; Lee, S.H.; Hur, K.Y.; Woo, S.Y.; Kim, S.W.; Kang, W.K. Statins and the risk of gastric cancer in diabetes patients. BMC Cancer 2012, 12, 596, doi:10.1186/1471-2407-12-596.
  18. Lim, T.; Lee, I.; Kim, J.; Kang, W.K. Synergistic Effect of Simvastatin Plus Radiation in Gastric Cancer and Colorectal Cancer: Implications of BIRC5 and Connective Tissue Growth Factor. J. Radiat. Oncol. Biol. Phys. 2015, 93, 316-325, doi:10.1016/j.ijrobp.2015.05.023.
  19. Goulitquer, S.; Croyal, M.; Lalande, J.; Royer, A.L.; Guitton, Y.; Arzur, D.; Durand, S.; Le Jossic-Corcos, C.; Bouchereau, A.; Potin, P., et al. Consequences of blunting the mevalonate pathway in cancer identified by a pluri-omics approach. Cell Death Dis. 2018, 9, 745, doi:10.1038/s41419-018-0761-0.
  20. Chushi, L.; Wei, W.; Kangkang, X.; Yongzeng, F.; Ning, X.; Xiaolei, C. HMGCR is up-regulated in gastric cancer and promotes the growth and migration of the cancer cells. Gene 2016, 587, 42-47, doi:10.1016/j.gene.2016.04.029.
  21. Follet, J.; Corcos, L.; Baffet, G.; Ezan, F.; Morel, F.; Simon, B.; Le Jossic-Corcos, C. The association of statins and taxanes: an efficient combination trigger of cancer cell apoptosis. J. Cancer 2012, 106, 685-692, doi:10.1038/bjc.2012.6.
  22. Cheung, K.S.; Chan, E.W.; Wong, A.Y.S.; Chen, L.; Seto, W.K.; Wong, I.C.K.; Leung, W.K. Statins Were Associated with a Reduced Gastric Cancer Risk in Patients with Eradicated Helicobacter Pylori Infection: A Territory-Wide Propensity Score Matched Study. Cancer Epidemiol. Biomarkers Prev. 2020, 29, 493-499, doi:10.1158/1055-9965.EPI-19-1044.
  23. Ma, Z.; Wang, W.; Jin, G.; Chu, P.; Li, H. Effect of statins on gastric cancer incidence: a meta-analysis of case control studies. Cancer Res. Ther. 2014, 10, 859-865, doi:10.4103/0973-1482.138218.
  24. Cho, M.H.; Yoo, T.G.; Jeong, S.M.; Shin, D.W. Association of Aspirin, Metformin, and Statin Use with Gastric Cancer Incidence and Mortality: A Nationwide Cohort Study. Cancer Prev. Res. (Phila) 2021, 14, 95-104, doi:10.1158/1940-6207.CAPR-20-0123.
  25. Graaf, M.R.; Beiderbeck, A.B.; Egberts, A.C.; Richel, D.J.; Guchelaar, H.J. The risk of cancer in users of statins. J Clin Oncol 2004, 22, 2388-2394, doi:10.1200/JCO.2004.02.027.
  26. Shimoyama, S. Statins and gastric cancer risk. Hepatogastroenterology 2011, 58, 1057-1061.
  27. Bujanda, L.; Rodriguez-Gonzalez, A.; Sarasqueta, C.; Eizaguirre, E.; Hijona, E.; Marin, J.J.; Perugorria, M.J.; Banales, J.M.; Cosme, A. Effect of pravastatin on the survival of patients with advanced gastric cancer. Oncotarget 2016, 7, 4379-4384, doi:10.18632/oncotarget.6777.
  28. Konings, I.R.; van der Gaast, A.; van der Wijk, L.J.; de Jongh, F.E.; Eskens, F.A.; Sleijfer, S. The addition of pravastatin to chemotherapy in advanced gastric carcinoma: a randomised phase II trial. J. Cancer. 2010, 46, 3200-3204, doi:10.1016/j.ejca.2010.07.036.
  29. Chiu, H.F.; Ho, S.C.; Chang, C.C.; Wu, T.N.; Yang, C.Y. Statins are associated with a reduced risk of gastric cancer: a population-based case-control study. J. Gastroenterol. 2011, 106, 2098-2103, doi:10.1038/ajg.2011.277.
  30. Spence, A.D.; Busby, J.; Hughes, C.M.; Johnston, B.T.; Coleman, H.G.; Cardwell, C.R. Statin use and survival in patients with gastric cancer in two independent population-based cohorts. Drug Saf. 2019, 28, 460-470, doi:10.1002/pds.4688.
  31. Karp, I.; Behlouli, H.; Lelorier, J.; Pilote, L. Statins and cancer risk. J. Med. 2008, 121, 302-309, doi:10.1016/j.amjmed.2007.12.011.
  32. Chung, H.; Kim, H.J.; Jung, H.C.; Lee, S.K.; Kim, S.G. Statins and metachronous recurrence after endoscopic resection of early gastric cancer: a nationwide Korean cohort study. Gastric Cancer 2020, 23, 659-666, doi:10.1007/s10120-020-01041-z.
  33. Kim, S.Y.; Min, C.; Oh, D.J.; Choi, H.G. Tobacco Smoking and Alcohol Consumption Are Related to Benign Parotid Tumor: A Nested Case-Control Study Using a National Health Screening Cohort. Exp. Otorhinolaryngol. 2019, 12, 412-419, doi:10.21053/ceo.2018.01774.
  34. Song, J.W.; Chung, K.C. Observational studies: cohort and case-control studies. Reconstr. Surg. 2010, 126, 2234-2242, doi:10.1097/PRS.0b013e3181f44abc.
  35. Orkaby, A.R.; Driver, J.A.; Ho, Y.L.; Lu, B.; Costa, L.; Honerlaw, J.; Galloway, A.; Vassy, J.L.; Forman, D.E.; Gaziano, J.M., et al. Association of Statin Use With All-Cause and Cardiovascular Mortality in US Veterans 75 Years and Older. JAMA 2020, 324, 68-78, doi:10.1001/jama.2020.7848.
  36. Kim, S.Y.; Min, C.; Oh, D.J.; Choi, H.G. Bidirectional Association Between GERD and Asthma: Two Longitudinal Follow-Up Studies Using a National Sample Cohort. Allergy Clin. Immunol. Pract. 2020, 8, 1005-1013 e1009, doi:10.1016/j.jaip.2019.10.043.
  37. Kim, S.Y.; Oh, D.J.; Park, B.; Choi, H.G. Bell's palsy and obesity, alcohol consumption and smoking: A nested case-control study using a national health screening cohort. Rep. 2020, 10, 4248, doi:10.1038/s41598-020-61240-7.
  38. Austin, P.C. Balance diagnostics for comparing the distribution of baseline covariates between treatment groups in propensity-score matched samples. Stat Med 2009, 28, 3083-3107, doi:10.1002/sim.3697.
  39. Li, F.; Thomas, L.E.; Li, F. Addressing Extreme Propensity Scores via the Overlap Weights. J. Epidemiol. 2019, 188, 250-257, doi:10.1093/aje/kwy201.
  40. Thomas, L.E.; Li, F.; Pencina, M.J. Overlap Weighting: A Propensity Score Method That Mimics Attributes of a Randomized Clinical Trial. JAMA 2020, 323, 2417-2418, doi:10.1001/jama.2020.7819.
  41. Li, F.; Morgan, K.L.; Zaslavsky, A.M. Balancing covariates via propensity score weighting. Am. Stat. Assoc. 2017, 113, 390–400.

  1. Comment: - check and correct English

Response: We apologize for the English errors. In fact, the manuscript has been edited by a professional English correction company, as described below. We will promptly get once again English correction via the fastest service.

Round 2

Reviewer 2 Report

The revised version of the presented manuscript meets all of my requirements. Authors made a lot of work to improve the quality of the manuscript. Authors gave comments and presented the answears for all of my questions.

The quality of the figures pointed in my review is stil low therefore in the final version it should be improved.

Then the manuscript can be accepted for publication.